# Advancing Event Forecasting through Massive Training of Large Language Models: Challenges, Solutions, and Broader Impacts

## Abstract

Many recent papers have studied the development of superforecaster-level event forecasting LLMs. While methodological problems with early studies cast doubt on the use of LLMs for event forecasting, recent studies with improved evaluation methods have shown that state-of-the-art LLMs are gradually reaching superforecaster-level performance, and reinforcement learning has also been reported to improve future forecasting. Additionally, the unprecedented success of recent reasoning models and Deep Research-style models suggests that technology capable of greatly improving forecasting performance has been developed. Therefore, based on these positive recent trends, we argue that the time is ripe for research on large-scale training of superforecaster-level event forecasting LLMs. We discuss two key research directions: training methods and data acquisition. For training, we first introduce three difficulties of LLM-based event forecasting training: noisiness-sparsity, knowledge cut-off, and simple reward structure problems. Then, we present related ideas to mitigate these problems: hypothetical event Bayesian networks, utilizing poorly-recalled and counterfactual events, and auxiliary reward signals. For data, we propose aggressive use of market, public, and crawling datasets to enable large-scale training and evaluation. Finally, we explain how these technical advances could enable AI to provide predictive intelligence to society in broader areas. This position paper presents promising specific paths and considerations for getting closer to superforecaster-level AI technology, aiming to call for researchers' interest in these directions.

## 1 Introduction

Future event forecasting is a task of predicting whether specific events will happen in the future, or what the probability of occurrence is, based on information up to a certain point in time (Jin et al., 2021; Zou et al., 2022; Halawi et al., 2024). For example, consider the question: "Today is December 31st, 2023. Will SpaceX successfully complete an orbital flight—reaching space and circling the Earth—before June 2024?" When solving this task with Large Language Models (LLMs), the widely used approach is Retrieve-Augmented Generation (RAG) (Lewis et al., 2020), whereby relevant news articles and related information are first retrieved, followed by reasoning processes that derive the final answer (Halawi et al., 2024). An important goal in the future event forecasting field is to make LLMs perform as well as top-level human forecasting experts or collective intelligence of general experts, that is, to make models reach superforecaster-level (Tetlock and Gardner, 2016; Karger et al., 2025; Liptay, 2024a).

Since ChatGPT was released (OpenAI, 2022), numerous studies have evaluated LLMs' event forecasting capabilities and compared them with human performance (Schoenegger et al., 2024; Hsieh et al., 2024). Initially, optimistic reports were shared that LLMs showed performance approaching superforecaster-level (Phan et al., 2024). However, subsequent analyses identified methodological issues including insufficient statistical significance, information leakage from data preceding the knowledge cut-off date, and contamination from

---

†Participated only in an advisory capacity.

post-resolution documents in search results, leading to criticism that LLMs' abilities were overestimated (Lopez-Lira et al., 2025; Bosse et al., 2024). These criticisms resulted in skepticism within the event forecasting community (Paleka et al., 2025a; Matthews, 2025).

However, we argue that recent studies provide positive signals for event forecasting. A recent study using more rigorous evaluation methods (Karger et al., 2025) reports that LLM performance in event forecasting is steadily improving with generational advances, and they are getting closer to superforecaster-level. Additionally, recent reasoning models like OpenAI o1 and o3 (OpenAI, 2024; 2025a) have shown improved performance compared to previous models (Hickman, 2025), and performance improvements through reinforcement learning (RL) have also been reported (Turtel et al., 2025a;b). Furthermore, the unprecedented success of reasoning model with tool-use like OpenAI's and Gemini's Deep Research (Citron, 2024; OpenAI, 2025b; Anthropic, 2025) suggests that technology capable of greatly improving forecasting performance has been developed.

Based on these recent positive trends, we argue that conditions are now favorable for research on large-scale training of event forecasting LLMs to approach superforecaster-level performance. This paper presents two key research directions for this purpose: training methodology (Section 4) and large-scale data acquisition (Section 5).

For training methodology, we first introduce three unique difficulties in LLM-based event forecasting training. First is the noisiness and sparsity problem, which is the difficulty in learning due to inherent uncertainty in event forecasting outcomes and the sparsity of similar events. Second is the knowledge cut-off problem, where it is difficult to train or evaluate event forecasting questions about knowledge that LLMs already know internally, greatly limiting usable training data. Third is the simple reward structure problem, where models can obtain rewards more easily than in other RL tasks without developing proper reasoning capabilities, hindering actual prediction ability improvement.

To mitigate these problems, we present several solutions. We provide theoretical grounds for various training label assignment strategies through hypothetical event Bayesian network modeling, introduce methods of utilizing poorly-recalled data and generating counterfactual events to tackle the knowledge cut-off problem, and discuss ways to solve the simple reward structure problem through auxiliary reward signals and subquestions.

For large-scale data acquisition, we point out that previous research mainly relied on prediction markets and propose aggressive use of three data categories: (1) market dataset - data available from prediction markets like Polymarket and Metaculus, (2) public dataset - structured data available from public databases like GDP and economic indicators, and (3) crawling dataset - unstructured data collected and processed from the web like news. Using these diverse data sources will enable large-scale training and fast evaluation cycles, promoting model performance improvement and development of generalized event forecasting capabilities.

Finally, we discuss the broad impacts these technical advances could have on society (Section 6). We examine promising applications, including expanding the scope of AI forecasting, AI-assisted trading systems, future simulation capabilities, and integrating probabilistic reasoning capabilities into general AI agents and AI scientists. We also analyze key challenges, including assessing prediction confidence, user interface design, self-fulfilling prediction effects, and vulnerability to malicious attacks.

This position paper provides a comprehensive review of event forecasting with LLMs, arguing that recent advances in LLM capabilities have created favorable conditions for large-scale training toward superforecaster-level AI systems. We identify and formalize the unique training challenges specific to event forecasting, propose methodological solutions to address these challenges, and develop strategies for performance enhancement through large-scale dataset expansion. In addition, we conduct a systematic analysis of the societal implications of event forecasting LLMs, examining both their potential for widespread adoption and associated risks.

## 2 Background

This section provides background on event forecasting using established terminology in the field (Table 1).

| Term | Example | Definition |
|------|---------|------------|
| Question | Today is Dec 12, 2023. Will SpaceX successfully launch and return a spacecraft from Earth orbit by 2024 June? | The question which asks about whether a specific event will happen by a certain time. |
| Question date | Dec 12, 2023 | The date that the question is asked. LLM must utilize the knowledge before this date to answer the question. Therefore, this must be the "event knowledge cut-off date" for the LLM. |
| Resolution date | Mar 14, 2024 | The date that the outcome of the event is determined. In other words, the date that the outcome is resolved. |
| Outcome | Yes | The outcome of the event known and finalized from the resolution date. It remains unknown before the resolution date, leaving the outcome unresolved. |
| LLM knowledge cut-off date | Nov 30, 2023 | The latest date of the knowledge an LLM is trained on. |

Table 1: Examples and explanations of major terms. The event example is referenced from (Polymarket, 2023).

## 2.1 Prediction market

A prediction market is a platform where users bet on whether specific events will occur. Each market within prediction market platforms corresponds to a specific event and provides market predictions for outcomes at different time periods for that event (Pratt et al., 2024). Market predictions represent the aggregated probability estimates from participants regarding whether a specific event will occur before resolution. Representative prediction markets include Polymarket, Metaculus, and Manifold Markets. Polymarket uses real money, whereas Metaculus and Manifold Markets use virtual currency. These platforms democratize future knowledge by providing reliable probability estimates for significant global events through various mechanisms (Williams, 2025a; Chen, 2022). For example, Polymarket achieved recognition for delivering more accurate predictions for the 2024 U.S. presidential election compared to expert analysts and political forecasting platforms (Jones, 2024).

Prediction markets play a significant role in the field of event forecasting AI. They are widely used as sources of both training and evaluation data for developing such systems. Moreover, matching the forecasting performance of prediction markets and scaling it to a wider range of prediction problems is one of the key motivations for event forecasting AI research.

## 2.2 Superforecaster

Superforecaster (Tetlock and Gardner, 2016) is a term referring to top forecasters who have exceptional talent in prediction compared to the general public. The criteria for superforecaster-level vary slightly across different literature and experimental settings, but recent studies define it as the prediction level of collective intelligence of forecasting experts (Karger et al., 2025; Liptay, 2024a). Forecasting experts refer to professionals hired as forecasters, while collective intelligence represents the aggregated predictions from multiple experts. Karger et al. (2025) asked the general public and forecasting experts a total of 200 questions, with randomly selected 20 questions each, and used their combined answers for comparison with models. Metaculus AI Benchmarking (Liptay, 2024a; Hickman, 2025) hosted by Metaculus is also an active challenge in this field. In this challenge, the performance of AI systems is compared using approximately 300 questions. Furthermore, the top-performing AI systems from previous competitions are combined into an ensemble, which is then evaluated against groups of approximately 10 expert forecasters using around 100 questions to assess the performance gap.

This definition clarifies that, contrary to the common first impression of the term "superforecaster," superforecaster-level AI does not mean a prophet who predicts everything perfectly. Superforecaster-level simply means the top level of human expert forecasters or the collective intelligence level of expert forecasters.

## 2.3 Benchmark

**Static benchmark**   ForecastQA (Jin et al., 2021), AutoCastQA (Zou et al., 2022), and AutoCast++ (Yan et al., 2024) are early major event forecasting benchmark studies. These benchmarks can be classified as static benchmarks because they consist of data from specific periods in the past that have already been resolved (Ye et al., 2024). The main challenge of static benchmarks is that data contamination can easily occur due to the nature of future prediction. If an LLM is developed in 2024 but the evaluation data contains questions about events that occurred in 2023, it cannot be used for evaluation because it may already be contaminated through training. Therefore, static benchmarks soon become outdated and cannot be used to evaluate the latest LLMs.

**Dynamic benchmark**   In contrast to static benchmarks, dynamic benchmarks continuously update their question sets and resolution information from the latest databases. Here, models answer unresolved questions posted in the database at specific times, and when these questions are resolved after a certain period, the model receives a score based on the resolution outcome. Since previous static benchmarks created reliability issues related to contamination, dynamic benchmarks are considered a major advancement. Recently, dynamic benchmarks and related challenges continue to be shared (Karger et al., 2025; Paleka et al., 2025b; Liptay, 2024a). Recent dynamic benchmark studies use market data from prediction markets as a key source for performance evaluation.

**Metric**   Benchmarks compare the error between the model's probability predictions and the actual resolved results using various metrics. The Brier score is a commonly used metric. The Brier score is defined as $(f - o)^2$, where $f \in [0, 1]$ is the probabilistic forecast and $o \in \{0, 1\}$ is the outcome after the event is resolved. Lower Brier scores indicate better performance, and a uniform prediction of 50% creates a Brier score of 0.25 (random baseline). Another metric is the logarithm score, defined as $o \log f + (1 - o) \log(1 - f)$. The logarithm score is more sensitive to extreme errors in probability estimates. Expected Calibration Error (ECE) is a metric that measures whether the actual outcome resolution probability of questions the model predicted with probability $p$ is close to $p$. Generally, specific interval bins (e.g., 5%) are set, and the absolute difference between the actual outcome resolution probabilities for prediction data within that range is used as the average metric. The concurrent position paper by Paleka et al. (2025a) addresses the difficulties in event forecasting evaluation and provides detailed explanations of these metrics along with comprehensive comparisons of their limitations.

## 2.4 Inference

**Information retrieval**   Information retrieval plays an important role in event forecasting systems (Hsieh et al., 2024). The study by Halawi et al. (2024) provides foundational research for current event forecasting and information retrieval pipeline design. In this study, they used an appropriate LLM-RAG-based pipeline to significantly improve prediction performance compared to cases without search. First, the LLM generates search queries related to the question, which are then used to conduct news searches with the search period restricted to information available before the question date. Then, the retrieved documents are reranked. Finally, answers are generated based on the organized documents. Additionally, Metaculus (Hickman, 2025) now provides an integrated information retrieval system with commercial LLM APIs to support forecasting research and practice.[1]

In practice, ensuring temporal integrity in document retrieval remains a significant challenge in event forecasting systems, particularly when the question date differs from the current time point. It is difficult to guarantee that retrieved documents do not contain information published after the question date, which can lead to data leakage and overestimated performance. Therefore, additional research and system development

---

[1]https://github.com/Metaculus/metac-bot-template/

on future information leakage are required. For example, Wildman et al. (2025) introduced a static benchmark with retrieved documents before the question date for each question using RetroSearch technique (Bosse et al., 2025). 20,000 documents are provided for each question on average. On the other hand, Turtel et al. (2025b) reported that they used a specific search API, Exa.ai API, which prevents information leakage.

**Ensemble**  Many forecasting studies employ LLM ensemble methods to enhance final performance (Halawi et al., 2024; Karger et al., 2025). This approach can involve generating multiple predictions from the same model using different prompts or ensembling predictions from multiple distinct models. Schoenegger et al. (2024) investigated ensemble effects using 12 models and compared their performance against human forecasters.

**Label type**  This paper explains event forecasting focusing on binary problems. Many event forecasting studies handle problems in binary classification form, or solve other types of problems by converting them to binary form. However, event forecasting can handle diverse problem types beyond binary classification. Multi-option, continuous (usually dates or numbers), entity-type open-ended (similar to multi-option but options are not given as choices), and sentence-type open-ended are problem types that can be handled in event forecasting (Wang et al., 2025). Most problem types can be relatively easily converted to binary problems. Multi-option or entity-type open-ended problems can be converted to binary problems for each individual option, and continuous problems can be converted to binary problems by dividing the range into appropriate intervals.

LLMs can generally perform well across all of these problem types. For example, when asked to infer probabilities of specific dates or numbers (continuous), LLMs can approximate the probability distribution over continuous values by providing probability estimates at regular intervals (e.g., 5% intervals from the 5th to 95th percentile). While recent work on training (Halawi et al., 2024; Turtel et al., 2025a;b) for event forecasting have focused on binary outcomes, future work could investigate whether direct training on multi-option or continuous predictions outperforms binarization approaches.

**Consistency**  Many reports have mentioned that LLMs exhibit poor consistency in probabilistic reasoning. The prediction that a candidate will lose by April and the prediction that they will lose after April should sum to 100%, but LLMs often do not (Liptay, 2024b). Lyu et al. (2025) discussed that LLMs have poor probabilistic consistency ability, and suggested the possibility that probability estimation performance could improve if this is handled well. The winner of Metaculus AI Benchmarking 4Q tackled the consistency problem well to achieve results that surpassed other AI systems (Hickman, 2025). The winner improved model predictions by having the model consider additional related options beyond the two choices presented in binary problems, thereby improving performance in the challenge. For example, if asked whether there would be the first negative GDP growth in the fall, they also asked about growth in winter and summer at the same time. Paleka et al. (2025b), which deeply discussed consistency, not only pointed out existing problems in the event forecasting field but also proposed related datasets. In the proposed dataset, they evaluate whether LLMs follow probabilistic conditions that should be mathematically satisfied for 10 different consistency rules, including negation, paraphrasing, and consequence.

## 3 Past and current state of event forecasting

**Evaluation problems in previous studies**  Following the emergence of ChatGPT and similar models, numerous reports emerged claiming that LLM-based systems achieved near-superforecaster performance, particularly throughout 2024 (Phan et al., 2024). However, subsequent analyses identified methodological flaws in these early optimistic analyses (Bosse et al., 2024; Paleka et al., 2025a). First, some studies drew excessive conclusions based on small samples that lacked sufficient statistical power to support their claims. Second, some studies erroneously used events that were resolved prior to the LLM's knowledge cut-off as evaluation instances, creating situations where models could simply recall memorized information (Lopez-Lira et al., 2025). Third, data contamination cases were also reported where documents from after the prediction resolution time were mixed into search results during web searches (Hendrycks and Mazeika, 2024).

These methodological issues led to criticism that studies systematically overestimated LLM capabilities (Bosse, 2023a). However, Matthews (2025) present a balanced assessment of both promising developments and ongoing challenges in event forecasting, explaining that while there have been limitations in recent academic progress, there are still reasons to pay attention to AI prediction technology development.

**Recent AI performance advances**   Indeed, positive trends are being observed in recent developments in the event forecasting field, according to recent studies using rigorous evaluation with dynamic benchmarks. The ForecastBench paper (Karger et al., 2025) created a dynamic benchmark and evaluated various LLM systems in summer 2024. They showed that while LLMs are still far from reaching superforecaster-level, LLM performance in event forecasting develops along with LLM performance improvements. Specifically, the authors highlighted strong correlations between event forecasting Brier scores and both (a) Chatbot Arena (Chiang et al., 2024) scores and (b) estimates of pretraining compute, implying that increases in general LLM performance directly affect improvements in event forecasting performance. Compared to early open-source models like GPT-3.5-Turbo and Llama-2-70B which had Brier scores exceeding 0.2, recent high-performance models like GPT-4o (Brier score 0.133 in the paper) and Claude-3.5-Sonnet (Brier score 0.122) have significantly narrowed the gap to Superforecaster AI (Brier score 0.096). This analysis also evaluated the collective intelligence of the general public, finding that when aggregating forecasts from general public participants, the median prediction achieved a Brier score of 0.121, similar to the best AI level. Generally, collective median predictions offset individual participants' biases and errors, performing much better than individual predictions; at least in the paper's benchmark, AI performance has likely already significantly surpassed average individual performance.

Furthermore, recent insightful reports on the Metaculus AI Benchmarking Series examine performance differences between evaluation experts and AI systems, sharing the trends that the latest models like OpenAI o1 and o3 (OpenAI, 2024; 2025a) consistently and significantly outperform previous models in these challenges (Liptay, 2024a; Hickman, 2025; Wilson and Bash, 2025; Williams, 2025b).

**Performance improvement through RL**   Recent achievements by Turtel et al. (2025a;b) regarding training are also noteworthy. These studies showed that reinforcement learning with verifiable rewards (RLVR) (Lambert et al., 2024; Guo et al., 2025) on outcomes can increase model performance. They conducted training and evaluation of the R1-14B model based on Polymarket datasets, showing that the R1-14B model with an original Brier score of 0.214 could reach OpenAI o1's 0.197 level through learning. Furthermore, they showed that additional data augmentation could further slightly improve performance, reduce algorithmic variance, and lower the model's overall ECE. Additionally, they showed in backtest experiments on Polymarket — experiments evaluating making money by betting on Polymarket's market in a virtual environment — that OpenAI o1 model and their trained algorithms could generate profits. They simulated trading by conducting trades when there were differences between the model's predicted values and market predicted values. While the long-term viability of LLM-based algorithmic trading in prediction markets requires further validation through real-world implementation, this result supports the argument that trained LLM models are getting closer to superforecaster-level AI.

**Deep Research**   Recent reasoning models with tool use like Deep Research (Citron, 2024; OpenAI, 2025b; Anthropic, 2025), o3, etc., also have the potential to be a major breakthrough for the event forecasting field. There are grounds to think their model structures could be suitable for event forecasting, especially when they are further trained with event forecasting-specific objective functions. They try various search and reasoning strategies on their own and attempt to solve problems by themselves, departing from the standardized prompt engineering-based or template-based reasoning used previously (Futuresearch, 2025). Furthermore, regression tests that infer current data values based on previous data are an important technique in event forecasting, and Deep research models have the capability to perform programming-based regression tests well (Liu et al., 2025). Therefore, using these models as a foundation model while incorporating event forecasting-specific training and inference strategies could yield substantial performance improvements.

**Proposal for large-scale training**   Based on these positive trends, we propose large-scale training for event forecasting LLM development. The current situation where model performance continues to improve and various training-related technologies are being developed suggests that opportunities have emerged to

close the gap to superforecaster-level performance through comprehensive large-scale training. We address two key research directions for large-scale training in the next two sections: training (Section 4) and dataset (Section 5).

Section 4 covers reward signal strategies and synthetic data generation methods to enhance efficiency within existing datasets, and Section 5 focuses on large-scale data collection. While these approaches can function independently, several algorithms from Section 4 create synergies with Section 5. In particular, solutions for the knowledge cut-off problem discussed in Section 4.2 enable models to effectively learn from historical patterns from the training instances before the knowledge cut-off, as discussed in Section 5.

## 4 Training algorithm improvements for future event forecasting

In this section, we introduce three difficulties of event forecasting tasks that other AI tasks do not have when it comes to model training. Then, for the purpose of understanding these difficulties and enabling people to think about additional research directions based on this, we introduce several training ideas to mitigate these difficulties.

The first training difficulty is the noisiness and sparsity problem of event forecasting outcomes (Kendall and Gal, 2017). For example, consider the problem of predicting the outcome of a US presidential election based on the initial situation. Since we can only make probabilistic inferences, not logical ones, based on information about the initial situation, the prediction label is noisy. Also, since presidential elections only occur once every four years, similar cases that can be used for training are sparse, making it difficult for models to learn sufficient patterns. We introduce the concept of hypothetical event Bayesian networks that can model this series of problems, and based on this, we discuss what kind of reward signals can be used for training.

The second training difficulty is the knowledge cut-off problem, which is the difficulty of training or evaluating event forecasting questions about knowledge that LLMs already know internally. This greatly reduces the amount of data that LLMs can train on. As one approach to this problem, we introduce utilizing events that LLMs do not recall well, such as comparative outcomes between two items. As another approach, we present the idea of training counterfactual events together so that models can focus more on search and reasoning.

The third training difficulty is the simple reward structure problem, where models can obtain rewards more easily than in other RL tasks without proper reasoning, hindering actual prediction ability improvement. To mitigate this problem, we propose auxiliary label training that can provide additional reward signals. One example of auxiliary labels is evaluating the consistency of reasoning, and another is having the model answer additional questions related to the main question.

### 4.1 Modeling hidden probability of future event forecasting

Let's say we want to train on whether "Today is Dec 12, 2023. Will SpaceX successfully launch and return a spacecraft from Earth orbit by 2024 June?" What should we use as the label for training and what training method should we use?

$$
\begin{array}{rccccc}
\text{Timeline:} & t_0 \text{ (Dec)} & \rightarrow & t_1 \text{ (Feb)} & \rightarrow & t_2 \text{ (Mar)} \\
\text{Information:} & S_0 \text{ (initial situation)} & \rightarrow & S_1 \text{ (intermediate update)} & \rightarrow & S_2 \text{ (final result)} \\
\text{Available:} & m_0 \text{ (market)} & \rightarrow & m_1 \text{ (market)} & \rightarrow & o \text{ (outcome)}
\end{array}
$$

When training models, the natural and intuitive label in event forecasting is the outcome that can be known after the event is resolved. For example, since SpaceX successfully completed orbital flight on March 14, 2024, the actual outcome is 1.0 (success). However, if the prediction probability of a training instance becomes 1.0 during the actual training process, it suggests that the LLM may not be conducting good search and reasoning. Since the purpose of training is to generalize search and reasoning abilities, such extreme predictions hinder the development of core abilities in event forecasting. From the perspective of assigning appropriate probabilities to training instances, market prediction obtained from prediction markets would be a good

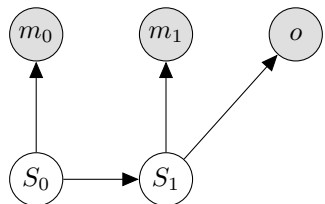

Figure 1: A hypothetical event Bayesian network. The state $S_0$ at time $t_0$ changes to $S_1$ at time $t_1$, and the probability of outcome $o$ changes. At time $t_0$, we get market prediction value $m_0$, and at time $t_1$, we get $m_1$.

estimate for that event through collective intelligence. However, if we train only on market predictions, it becomes difficult to create event forecasting prediction models that surpass market predictions.

In this subsection, we first discuss the level of noisiness and sparsity of this problem to explain how this dilemma makes event forecasting problems difficult from a machine learning perspective. Then we introduce hypothetical event Bayesian networks that can theoretically understand this problem. Based on this conceptual analysis, we discuss how each approach to assigning labels in event forecasting can be explained and what label construction strategies and their variations can be used.

### 4.1.1 Noisiness and sparsity

Event forecasting labels have two difficulties related to uncertainty built into them: the first is noisiness, and the second is sparsity. Regarding noisiness, consider the problem of predicting which candidate will win the 2024 US presidential election based on information available during the early stages of a competitive electoral campaign. No matter how much a smart expert analyzes available information from the early stages of the election, they cannot know for certain who will become president or what the probability is in percent. This means that the outcomes or market predictions we can use are inherently uncertain and noisy in explaining events.

Regarding sparsity, US presidential elections are events that happen once every four years, making them very sparse events. Presidential elections exist for each country and each cycle, so we might be able to achieve generalization through this, but the context of presidential elections differs for each event, and it is difficult to generalize from one case to other cases. Although event frequency varies by domain, this illustrates the inherent sparsity of comparable training instances in event forecasting problems. From a traditional ML perspective, the former (noisiness) corresponds to aleatoric uncertainty, and the latter (sparsity) is related to epistemic uncertainty: events for which we have less data will have higher associated epistemic uncertainty (Kendall and Gal, 2017). The combination of these factors means that event forecasting faces higher uncertainty than many other ML problems.

### 4.1.2 Hypothetical event Bayesian networks

We posed the question of what should be used as labels for event forecasting problems at the beginning of this subsection. As a conceptual framework for understanding this problem, let's construct a simple hypothetical event Bayesian network. Using the SpaceX example mentioned earlier, we model a situation where there is intermediate information in February ($t_1$)—the success or failure of initial tests and reports—between the question date $t_0$ in December and the resolution date $t_2$ in March. In our model, we define three core probabilities: $\alpha$: final success probability when initial test fails (negative), $\beta$: final success probability when initial test succeeds (positive), $\pi$: probability that initial test succeeds (positive). Figure 1 is a diagram of our hypothetical event Bayesian network. Equations are as follows:

$$\alpha = P(o = 1|S_1 = \text{negative})$$
$$\beta = P(o = 1|S_1 = \text{positive})$$
$$\pi = P(S_1 = \text{positive}|S_0 = \text{initial})$$

We seek to estimate an accurate probability for this question. Since the question date is $t_0$, we need to estimate the probability that the outcome will occur at the $S_0$ point. We refer to this probability we are interested in as "hidden probability." The hidden probability $P_{hidden}$ at the question date is as follows:

$$P_{hidden} = P(o = 1|S_0 = \text{initial}) = (1 - \pi)\alpha + \pi\beta$$

To estimate $P_{hidden}$, we cannot observe these underlying probabilities $\alpha$, $\beta$, and $\pi$ directly, and instead need to infer these probabilities. One good estimate for $P_{hidden}$ is a market prediction obtained through the collective intelligence of human forecasters. Market predictions $m_0$ and $m_1$ are estimates of $P(o = 1|S_0)$ and $P(o = 1|S_1)$, respectively, and we can assume they estimate values based on noisy observation of the actual parameters $\alpha$, $\beta$, and $\pi$.

In this hypothetical modeling, we can analyze how the accuracy of $P_{hidden}$ estimation differs according to noise level and $N$ by sampling $m_0$, $m_1$, $o$ $N$ times each (i.e., we simulate $N$ similar events, yielding $\{m_{0,n}, m_{1,n}, o_n\}_{n=1,\cdots,N}$). $N$ represents the number of hypothetical simulation trials. For example, in each trial, $S_1$ is sampled as positive with probability $\pi$. We can then compute separate estimates of $P_{hidden}$ by averaging the $m_0$ values, $m_1$ values, and $o$ values, respectively (i.e., $\frac{1}{N}\sum_{n=1}^{N} m_{0,n}$, $\frac{1}{N}\sum_{n=1}^{N} m_{1,n}$, and $\frac{1}{N}\sum_{n=1}^{N} o_n$).

How does the relative accuracy of each estimate differ at different $N$ values? First, let's think about the case where $N = 1$. Since $o$ has values of 0.0 or 1.0, there will be a considerable gap from $P_{hidden}$. In this case, $m_0$ is likely to be a much stronger estimate than $o$. However, let's think about the case where $N$ is sufficiently large. The estimation of $m_0$ contains prediction uncertainty noise, and this noise will slow convergence to $P_{hidden}$. In this case, $o$ will be a better estimate compared to $m_0$.

Can we use $m_1$ to estimate $P_{hidden}$? Market predictions at the intermediate time $m_1$ occupy a middle ground between $m_0$ and $o$. Unlike $m_0$, which must account for uncertainty about whether the intermediate state $S_1$ will be positive or negative, $m_1$ is made after this uncertainty is resolved. However, unlike the final outcome $o$, $m_1$ still reflects the collective judgment of forecasters rather than a binary result. This positioning can make $m_1$ a superior estimate of $P_{hidden}$ under certain conditions, particularly when there is significant uncertainty in the transition from $S_0$ to $S_1$ (high estimation error for $\pi$) and when we have a moderate number of observations $N$. We provide detailed assumptions and simulation results demonstrating these trade-offs in Appendix A.

Based on the hypothetical settings discussed above, let's now consider real scenarios of event forecasting tasks and address the question of whether to use outcome $o$ or market prediction $m_0$. Applying the hypothetical event Bayesian network framework, for data where there are few similar problems in the training data (small $N$), it would be good to use market prediction $m_0$ at question date time $t_0$, and for cases where there are many similar problems (large $N$), it would be good to use outcome $o$ at resolution date time $t_2$. In practice, using both together could also be a way to get better estimates in some cases. According to the simulation in Appendix A, interestingly, there are also cases where using the market prediction value $m_1$ at time $t_1$ between $t_0$ and $t_2$ as a label for questions at time $t_0$ has advantages, and $m_1$ does not function merely as an approximation of $o$. The next subsubsection examines label assignment strategies that can be used based on this discussion.

### 4.1.3 Outcome as a reward signal

The most direct method in event forecasting is using outcomes as rewards in RL. For example, we can use the negative Brier score between the probability predicted by the model and the actual outcome as a reward term (Turtel et al., 2025b), or give higher rewards to predictions closer to outcomes (Turtel et al., 2025a). Turtel et al. (2025b;a) achieved 5-10% Brier score improvement with training on outcomes; using outcomes in situations where an appropriately scaled dataset is secured can be a powerful baseline and practical approach.

However, outcome-based training has the risk of models making extreme predictions approaching 0% or 100% after sufficient training epochs (Turtel et al., 2025b). More importantly, extreme predictions approaching 0% or 100% are likely signals that the LLM is not actually performing proper search and reasoning. The ultimate goal of event forecasting training is to learn good search and reasoning abilities and transfer them to test

inference time, but extreme predictions hinder the development of these core abilities. Various regularization methods, such as early stopping, can be used to prevent this.

### 4.1.4 Market prediction as a reward signal

Prediction markets serve as effective platforms for obtaining expert estimates of hidden probability $P_{hidden}$ in a scalable way. Each market prediction is a good estimate for that event, and from the hypothetical event Bayesian network perspective discussed earlier, it is an advantageous estimate to use when similar events occur infrequently.

However, if we train only with market prediction, it becomes difficult to create event forecasting prediction models that surpass market prediction. Therefore, market prediction should be used as a reward signal in parallel with outcomes. In the research by Halawi et al. (2024), they used a specific interval between market prediction and outcome as the ground truth label. Specifically, they defined a 15% interval from the market prediction value in the direction of the actual outcome, and treated predictions within this range as correct answers. For example, if market prediction is 60% and outcome is 100%, they treated model predictions between 60% and 75% as correct. Additionally, various other methods for using market prediction and outcomes together in training can be considered. One method involves adding a KL-divergence term between model predictions and market predictions to the objective function, alongside the existing outcome-based reward signal. In another method, market predictions can be included directly as model inputs.

The reliability of each market is also an important consideration. For example, markets with limited participation may exhibit higher noise levels (Bosse, 2023b). In (Halawi et al., 2024), they filtered out market data with few participants. Market reliability can be incorporated based on the specific approach used for market prediction. When using KL-divergence, the weight on the KL-divergence can be reduced for training instances corresponding to markets with few participants. When inputting market prediction values, related information about reliability can also be added together.

Market predictions are available only when a prediction market exists for the specific event in question. When training on datasets containing both instances with available market predictions and instances without market data, reward signals derived from market predictions would be applied solely to those training instances where market prediction values are available.

### 4.1.5 Using prediction after the question time

A considerable number of event forecasting questions in the world may not have outcomes determined yet at the current time point when we are training. Also, these forecasting questions may not have clearly defined resolution conditions, or there may be no pipeline to automatically extract outcomes for data construction because events are not registered in markets or databases. Can we train from such data where outcomes are not clearly obtained?

If we can obtain market prediction $m_1$ at time point $t_1$ after question date $t_0$, we can obtain reward signals from such data as well. There are two perspectives we can think about regarding the validity of using $m_1$. The first is thinking of $m_1$ as a more accurate approximation of $o$ than $m_0$. Especially if the value of $m_1$ is close to 0.0 or 1.0, and if $m_1$'s estimation of $o$ is close to unbiased (if the model's ECE is low), we can statistically say that $o$ predicts well at the probability that $m_1$ suggests. From that perspective, when we get $m_1$ that is sufficiently different from $m_0$ and close to 0.0 and 1.0, we can use $m_1$ for training. The statistical validity can be established through empirical data analysis. The second is understanding $m_1$ as an approximate estimate of $P_{hidden}$ where uncertainty before $t_1$ has been resolved, as discussed in Section 4.1.2 earlier. In this interpretation, $m_1$ has meaning beyond being an estimate of $o$, and has the potential to provide better signals than $o$ in certain scenarios.

Next, we discuss whether the model's prediction $q_1$ at time $t_1$ can be used for model prediction where the question date is $t_0$. When market prediction $m_1$ at time $t_1$ cannot be obtained because the prediction market does not handle that event, the use of model prediction $q_1$ becomes a consideration. Our hypothetical Bayesian network framework extends naturally from market predictions $(m_0, m_1)$ to model predictions $(q_0, q_1)$. Critically, while $q_0$ offers no new information for training purposes, $q_1$ can leverage the temporal

information gain between $t_0$ and $t_1$, making it a viable training signal similar to $m_1$. The applicability of $q_1$ can be evaluated using similar criteria as those for $m_1$. Regarding the effectiveness of using $q_1$, we need to additionally consider the performance of the predicting LLM itself. If the prediction model's predictions are not sufficiently accurate and have high noise, it may be difficult to use $q_1$ due to the high noise. Even when asking LLMs at time points after resolution date $t_2$, if the LLM's search/RAG performance or fact checking performance is not 100%, the LLM may not achieve 100% accuracy.

Notably, the model's prediction $q_1$ can be applied to forecasting questions where outcome resolution conditions are not rigorously defined. For instance, this value can be used for questions with ambiguous resolution conditions like "Did ChatGPT released in 2023 have a positive impact on English education?" or "Will COVID vaccines effectively prevent COVID?" This approach extends event forecasting from questions with clear resolution criteria to more general questions about the future by leveraging the information gap between past ($S_0$) and present ($S_1$) states as training signals.

### 4.2 Training from data before knowledge cut-off

In this part, we discuss the knowledge cut-off problem and introduce ideas to mitigate this problem, including (1) using events that LLMs do not remember well and (2) training counterfactual events together.

#### 4.2.1 Knowledge cut-off problem

When evaluating event forecasting problems using data from before the knowledge cut-off period, models can provide accurate responses by relying on their internal knowledge without employing search and reasoning capabilities. Since these memorization-based responses do not generalize to questions where the model has not observed the outcomes during training, data from before the knowledge cut-off loses its value for evaluation purposes. The same limitation applies to training. Models have reduced motivation to acquire additional information through search when they already possess the relevant knowledge internally, and similarly lack motivation to engage in reasoning processes for answers they have already memorized. Consequently, neither search capabilities nor reasoning abilities are enhanced through this training approach.

If we use past models with earlier knowledge cut-offs for event forecasting training, we can utilize more training data compared to the latest models. However, there are two major disadvantages to using past models. First, since the latest models outperform past models, we cannot leverage the capability improvements from LLM development (Karger et al., 2025). Second, the latest models possess the most recent knowledge. Models with more up-to-date knowledge will make better future predictions for questions that require recent trends as context, even without searching for that specific knowledge.

Whether training can be conducted using data from before the knowledge cut-off is an important question. First, since substantially more data exists before the knowledge cut-off than after, the feasibility of utilizing pre-cutoff data determines the scale of available training data. Another issue is that if training data is constructed within a limited period after the knowledge cut-off, we end up training on data that exhibits biases specific to that particular period. For example, consider predicting numerous economic indicators during an economic boom period. Such a prediction model may not perform adequately during economic downturns.

#### 4.2.2 Using events LLMs poorly recall

One approach to addressing the knowledge cut-off problem is to use events that happened in the past but that LLMs cannot answer well through memorization as training data. We can explore what knowledge LLMs do not answer well through memorization by asking LLMs questions about various domains and use this approach to identify suitable training data.

Types of information that LLMs do not memorize well are cases where LLMs know individual facts but do not memorize the relationships or comparison results between them. Wen et al. (2025) created an event forecasting task about which of two research ideas would show better performance on benchmarks. Existing LLM agents without special training showed only random baseline level (50%) performance in the above prediction task comparing the relative performance of two ideas, even though they had knowledge about

individual papers. In that paper, they improved performance by training on 7,000 paper idea pairs based on open-source models. The trained model was able to achieve 77% accuracy forecasting performance on 1,500 pairs after the knowledge cut-off by retrieving related past papers and knowledge.

If we apply this approach of predicting comparison results of two indicators from the same period to data from more domains, we should be able to create data that can be used for training to increase event forecasting performance, even though it is data from before the knowledge cut-off date. For example, we can consider (1) future relative market performance of products released at the same time, (2) comparison of specific future indicators of companies with similar backgrounds, and (3) comparison of future citation counts of papers presented at academic conferences.

However, it remains an open empirical question whether training on such poorly-recalled historical events improves performance on general event forecasting tasks. Validating this transfer represents an important direction for future research.

### 4.2.3 Putting counterfactual events

Another approach is to use counterfactual events to make LLMs utilize retrieved knowledge for reasoning and probability prediction even when training from data before the knowledge cut-off. In this idea, we use counterfactual events that have outcomes opposite to the actual events that happened in the past. The key insight of this approach is that since LLMs must reason based on retrieved information even in counterfactual scenarios, they develop actual reasoning abilities rather than simple memorization.

Related research in the question answering field that used counterfactual event-based retrieved knowledge utilization can be referenced (Paranjape et al., 2022). For example, Neeman et al. (2023) showed that training on counterfactual knowledge can increase models' search grounding and reduce hallucination. In this research, counterfactual documents were constructed by modifying the correct answer entity in search documents.

Specific ideas for creating counterfactual events are as follows. First, we need to create fictional search documents that support counterfactual events. We can make LLMs generate fictional news articles reflecting counterfactual outcomes and use them as if they were search results. For example, we can create a counterfactual event of "SpaceX Starship 3rd launch failure in March 2024," which is the opposite outcome of "SpaceX Starship 3rd launch success in March 2024," and create related fictional news articles. Here is an example of an implementable pipeline:

**Step 1: Base event selection and counterfactual divergence point setting** For each base event, we set a specific time point (divergence date) where actual events and counterfactual events split. We use actual search documents until this point, and from then on, we use generated counterfactual documents. To determine an appropriate divergence date, we can query the model about the probability of that event occurring at various past time points and select a time point where uncertainty is appropriately high. For example, if the model predicts a 60% success probability for an event at a January 2024 time point, this time point can be designated as a candidate for the divergence point. Then, based on the model's predicted outcome prediction probability (60%) at the divergence date for the base event, we sample the base event and divergence date pair from the entire set. This is to minimize the level of distortion of overall event probabilities when adding counterfactual events to training.

**Step 2: Time-ordered document generation** We generate search documents in chronological order based on counterfactual scenarios. Generated counterfactual documents cover events occurring after the divergence date (e.g., "engine test anomaly signals" reported in the pre-launch period). When generating documents, we match the style and format similar to actual news.

**Step 3: Counterfactual event construction** We construct the entire search results for actual forecasting questions. For periods preceding the divergence date, we use actual search documents, while for periods following the divergence date, we use counterfactual documents and provide them as search results. The time of the question date is set between the divergence date and resolution date.

For quality control, we need consistency checks of generated documents, reality verification, and style similarity confirmation with authentic news sources. Additionally, we can iteratively adjust the optimal counterfactual event ratio through systematic experimentation.

### 4.3 The simple reward structure problem

In this subsection, we first discuss the simple reward structure problem, where models can easily obtain rewards without proper reasoning in RL situations, and then introduce ideas to reduce this problem by providing auxiliary labels besides outcomes (Jaderberg et al., 2017).

#### 4.3.1 Characteristics of simple reward structure

Like other RL tasks, preventing overfitting is important in event forecasting. However, overfitting in event forecasting is distinctive due to its simple reward structure.

This simple reward structure problem becomes apparent when compared to other major RLVR tasks, such as mathematical or coding problems. In event forecasting, models can obtain rewards without proper justification—they can generate predictions closer to 0% or 100% and receive higher rewards regardless of whether their reasoning or information search supports such confidence. This contrasts with mathematical and coding tasks, where obtaining correct answers without sound intermediate reasoning is nearly impossible.

The noisiness and sparsity problems discussed in Section 4.1 make this problem worse. First, unlike mathematical or coding tasks where memorized answers to training examples are typically correct on identical test cases, event forecasting aims to learn $P_{hidden}$ values (e.g., 50% for coin flips) between 0% and 100%, rather than binary outcome. Consequently, when models overfit by making extreme predictions (0% or 100%), they fail to capture the true underlying probability distributions that minimize Brier score, even when the test case is identical to the training case. Second, while training with sufficiently large datasets could mitigate overfitting, event forecasting scenarios are inherently sparse.

Therefore, the simple reward structure readily facilitates overfitting in event forecasting. When the model relies on extreme predictions in situations where search-based evidence is insufficient to justify confidence, the model loses incentive to develop sophisticated search and reasoning capabilities.

#### 4.3.2 Evaluation on reasoning

To address this problem, we propose incorporating evaluation of reasoning processes in addition to outcome-based rewards. Since we are interested in improving the model's search and reasoning abilities themselves, including such auxiliary rewards is a natural approach. As a simple example, in the work of Turtel et al. (2025b), to train the Deepseek R1 model for English event forecasting, they applied penalties when Chinese text appeared in the reasoning process, treating this as an indicator of poor reasoning quality.

Further research is required to identify effective methods for creating reward signals that accurately assess reasoning quality. For instance, analogous to evaluation methods in general LLM alignment research, we can employ judge LLMs to assess reasoning appropriateness (Kim et al., 2025). Following standard practices in LLM reinforcement learning (Ouyang et al., 2022), we can enhance judge LLMs' reliability by collecting expert annotations comparing pairs of reasoning responses for event forecasting, then training a reward model based on these preferences.

#### 4.3.3 Asking subquestions

As another method for generating auxiliary rewards, we propose incorporating predictions on related questions that share underlying causal factors with the main forecasting task (Zhou et al., 2023). The core intuition is that genuine predictive understanding should manifest as coherent predictions across multiple related events, rather than isolated correct guesses.

Building on the consistency benefits demonstrated in Section 2.4, we extend beyond reformulated versions of the same question to semantically related questions that provide additional training signals. For example, when predicting which candidate will win a presidential election, related questions might include "Will

Candidate X lead in October polls?" or "Will candidates X and Y form a coalition or alliance?" These questions share underlying factors with the main prediction while providing independent evidence of the model's reasoning quality.

LLMs can be used to generate relevant subquestions. For example, subquestions can be generated using information available at the question date, focusing on factors that appear closely related to the main question. Alternatively, subquestions can be generated using hindsight from the resolution date, focusing on events that ultimately influenced the actual outcome. LLMs can also evaluate subquestion resolutions. Fact-checking models can be employed to determine outcomes and assess whether subquestions should contribute to auxiliary rewards.

## 5   Dataset construction for large-scale training for future event forecasting

In this section, we discuss that using more data for training than in previous research can improve event forecasting performance. The importance of data scale and diversity in event forecasting is demonstrated by several previous studies (Kaplan et al., 2020; Edwards et al., 2024). Early research (ForecastQA) showed a scaling law where test accuracy increased log-linearly as the dataset size increased (Jin et al., 2021). Recent work by Turtel et al. (2025a;b), which trained using a large amount of Polymarket data, shows that training performance increases when using many available datasets in event forecasting. They achieved performance improvements without the specialized data quality filtering used in previous research (Halawi et al., 2024). Turtel et al. (2025b) also showed that there were additional performance improvements through data augmentation, which implies the need for larger datasets.

We explain three data categories that can be used for large-scale training: market, public, and crawling datasets. Based on this explanation, we propose using larger and more diverse datasets for training than what was used in existing research. Currently, the event forecasting field is mainly focused on training with market datasets that are easy to receive real-time updated data. However, public datasets and crawling datasets that were mainly used only for evaluation can also be used for training. By combining multiple prediction markets with public databases and web-crawled content, training datasets could expand from the current 10,000 instances (Turtel et al., 2025a;b) to over 100,000 instances. The scale of this dataset can be further expanded by leveraging the scope of trainable questions (Sections 4.1.5 and 4.3.3), extending the temporal span of dataset resolution dates (Section 4.2), and incorporating methodological advances from research in other dataset domains. If there is enough data, we can distinguish between extensive data for further RL pretraining (Dong et al., 2025) and curated high-quality data for instruction learning. We discuss the possibilities and considerations for constructing and utilizing these large-scale datasets.[2]

### 5.1   Market dataset

**Market datasets and expansion strategies**   Market dataset refers to datasets based on market predictions traded on market platforms like prediction markets. As explained in Section 2.1, prediction markets and their associated datasets play a central role in event forecasting. Market datasets are already being used as key components in event forecasting training research, but there is still room to expand this on a larger scale to secure more datasets. Examining existing research, Halawi et al. (2024) constructed datasets from multiple sources but selected only a limited number of markets to create a total of 3,700 training examples, and Turtel et al. (2025a;b) used 10,000 examples but used only Polymarket datasets.

**Dataset quality**   However, in such large-scale expansion, the inevitable question raised is dataset quality. There is a trade-off between data quantity and quality, depending on how quality is defined and what filtering standards are applied. Below, we discuss this trade-off based on examples from previous research.

A study by Halawi et al. (2024) belongs to event forecasting research focused on data quality. In this research, two quality criteria were defined and corresponding data filtering was conducted. The first quality criterion is market trading volume. This research viewed markets with insufficient numbers of betting participants as low-quality data and excluded them from training data. In particular, since this research conducted training

---

[2]Use of the recommended datasets/data sources is subject to any applicable terms and laws.

centered on market prediction rather than outcomes, using this filtering can be seen as an appropriate criterion for that research. The second quality criterion is the objectivity of questions. In this research, they used LLMs to judge whether each question was subjective and filtered out subjective questions. Examples of subjective questions are "Will I finish my homework tonight?" or "My coin toss: heads or tails?". Manifold markets hosts numerous markets (questions); however many involve subjective questions like coin toss. Indeed, from a massive-training perspective for future prediction, subjective questions such as coin toss probabilities should also be learned as targets with appropriate probabilities (e.g., 50% for fair coin tosses) (Section 4.1.2). However, unlike other non-subjective questions, subjective events have more noise in general and make training less effective from a learning and reasoning perspective.

Meanwhile, more recent research by Turtel et al. (2025a;b) showed improved performance compared to previous research without specifically applying these quality criteria. These results imply that large-scale data can yield meaningful performance improvements even with relaxed data quality thresholds.

**Discussion**   Recent research trends in market dataset usage focus on performance improvement through large-scale data utilization rather than strict quality criteria. Therefore, this supports our position of proposing large-scale data utilization. However, systematic ablation studies are needed to find the optimal balance between data quality and quantity, and we expect that such studies will establish stable and scalable dataset construction strategies.

## 5.2   Public dataset

In this subsection, we explain training on public datasets. Public datasets refer to structured datasets defined in database form on the web. The scope is diverse, encompassing data such as quarterly UK GDP figures or Amazon review scores. This diverse scope enables a correspondingly extensive range of predictable questions. Public datasets have primarily been used for evaluation in dynamic benchmarks rather than as training data. We present several examples of public datasets, advocate for their use in training, and discuss key considerations for their implementation.

**Public dataset in ForecastBench**   We explain examples of public datasets based on the public dataset-based dynamic benchmark built in ForecastBench (Karger et al., 2025). They constructed a dynamic benchmark using 2,000 market dataset events and 4,000 public dataset events. A considerable part of the public datasets they used came from ACLED (ACLED, 2025), a geopolitical database dealing with world conflicts, comprising approximately 3,000 events. However, other public datasets that ForecastBench used only partially—DBnomics (DBnomics, 2025) and FRED (Federal Reserve Economic Data) (FRED, 2025)—are also noteworthy. DBnomics, a global economic database, reports on its homepage that it contains 30K datasets and 1B time series data provided by nearly 100 providers. FRED is a US-centered economic database, which also contains nearly 1B time series. ForecastBench used only a small selected part of these two data sources, and from DBnomics, they used only questions about weather information. However, since these databases have substantial time series, more data can be selected and used. Considering the scale of this data, conceptually, hundreds of thousands of instances or more can be systematically extracted even within specific time periods.

**Other public datasets**   Public databases, by definition, provide well-structured data in web database form, and various databases can be chosen based on the target domain of the model. For example, in the health field, WHO's Global Health Observatory data (WHO, 2025) or CDC's public health statistics (CDC, 2025) are available, and in the environmental field, NASA's climate data (NASA, 2025) or NOAA's weather data (NOAA, 2025) are available. Metaculus provides examples and uses of databases that human forecasters can use for analysis.[3]

**Inter-event correlation**   In large-scale expansion of public datasets, the important question raised is the correlation problem. The first correlation to consider is the correlation between events. If there are many datasets with high correlation, the i.i.d. assumption is violated, preventing the model from learning diverse information proportional to the data volume. Some public datasets exhibit this tendency. For example, many

---

[3]https://www.metaculus.com/help/prediction-resources/

economic indicators tend to follow similar patterns depending on whether the global economy is good or not. As a mitigation measure, by evaluating inter-event correlations and sampling data to maintain low inter-correlation across the entire dataset, learning efficiency relative to data volume can be preserved.

**Temporal correlation** The second correlation to consider is correlation within specific time periods. This addresses scenarios involving temporally localized patterns within specific periods, and these trends do not generalize beyond those periods. Taking economic indicators as an example again, patterns learned only during boom periods cannot generalize to patterns during recession periods. This poses a significant challenge in event forecasting where knowledge cut-off problems exist. If training is conducted only with the latest data for data groups with strong time-specific patterns, proper learning from past information becomes challenging. Therefore, for using such data, the ideas mentioned as examples in Section 4.2 for mitigating knowledge cut-off are essential. Through proper incorporation of pre-cutoff training data, the value of public datasets can be greatly enhanced.

**Discussion** For the effective use of public datasets, (1) systematic discovery and integration of domain-specific databases, (2) ensuring data diversity through correlation-based sampling strategies, and (3) applying learning methodologies that mitigate time-specific biases are necessary. The approach of using public dataset will enable supplementation of the current market dataset-centered learning paradigm and the development of more generalized event forecasting models.

## 5.3 Crawling dataset

In this subsection, we explain the concept of crawling datasets and training methods. Datasets can be constructed through processing web-based sources such as Wikipedia or news articles, which we term crawling datasets. We first explain several sources and related research cases that can be used for crawling datasets. Then, we discuss methods and difficulties for utilizing crawling datasets as training data, and encourage future research related to the usage of crawling datasets in training.

**Wikipedia** Wikipedia, which systematically organizes and updates new information, is a suitable source for creating prediction problems through data processing. For example, Wang et al. (2025) created questions and answers by crawling Wikipedia to make evaluation datasets. The authors built a pipeline that automatically creates data without human intervention. The data includes hundreds of thousands of questions spanning from 1950 to the present.

**News** Using LLMs to extract forecasting questions or outcomes from news articles is an effective method for building crawling datasets. Guan et al. (2024) provide valuable insights into methods for creating event forecasting datasets from news sources. This work constructs static event forecasting benchmarks by leveraging topical data from various news articles and forum posts. However, in this research, LLMs initially generate data, which humans subsequently review to finalize the evaluation set.

**arXiv** Event forecasting questions can also be created about arXiv paper data. As of 2025, approximately 20,000 new papers are published monthly.[4] By extending the framework of Lewis et al. (2021) that automatically generates QA questions, datasets can be created for forecasting tasks that predict information about specific papers without access to those papers themselves, using only historical research patterns. By creating questions about which papers will appear or gain prominence in the future, model capabilities for predicting future research trends can be strengthened. A relevant recent example is research on forecasting which ideas will achieve strong performance on specific AI benchmarks (Wen et al., 2025). In this research, they trained on 7,000 paper-idea pairs from before the knowledge cut-off to predict which of two ideas would perform better, and based on this, they predicted 1,500 pairs after the knowledge cut-off. The trained system achieved 77% accuracy by retrieving past papers and generalizing well to future ideas, achieving substantial performance gains compared to the baseline.

---

[4]https://arxiv.org/stats/monthly_submissions

**Other crawling sources**  While crawling datasets require domain-specific approaches that make field-by-field creation challenging, their possibilities are numerous. Depending on the target domains where improved model performance is desired, relevant datasets can be constructed accordingly. For example, to enhance models for predicting public opinion trends, opinion polling datasets can be gathered and incorporated into training data. If training incorporates publicly available individual blogs, personalized event predictions can be generated based on individual user histories.

**Event predictability validation**  The challenge in constructing crawling datasets lies in collecting questions that enable probabilistic predictions about future events through information retrieval. Therefore, when working backward from known outcomes to generate questions, one must verify whether the event is predictable from past information, and the appropriate timing when adequate information to predict the event becomes available must be identified. Dataset selection and question timing can be determined by assessing whether questions are too difficult or too easy by asking models to predict events using only knowledge available at past time points. When model predictions at specific time points show improved accuracy compared to earlier predictions while still maintaining meaningful uncertainty about the outcome, that time point can be designated as the question date.

**Automated pipeline reliability**  Like market datasets and public datasets, crawling datasets also need to be created in an automated way from incoming news or web data to achieve scalability. Thus, one of the main challenges for crawling datasets is the noise in questions and answers that arises from automated generation processes. While human post-review is possible, continuously reviewing all newly generated data raises a scalability issue. This automation-quality trade-off is related to the problem of using model predictions as labels discussed in Section 4.1.5. For training usage of crawling datasets, follow-up research on the accuracy of datasets created in automated ways and the impact that inaccurate datasets from various domains have on overall training is needed.

**Discussion**  Crawling datasets have the potential to substantially expand the scale and diversity of event forecasting training data to complement market datasets and public datasets. However, data quality management and ensuring the stability of automated generation pipelines remain key challenges. Key research directions include (1) developing domain-specific crawling strategies, (2) building automated data quality evaluation systems, and (3) applying noise-robust learning methodologies to improve the practical utility of crawling datasets.

## 5.4   Fast model evaluation with massive dynamic benchmarks

We explained methods for collecting large amounts of data for training in previous subsections, but the same methods can be applied to evaluation data collection. When collecting event data, data with resolved outcomes will mostly be used as training or validation data, while unresolved questions can be used as dynamic benchmark data.

**Constructing dataset for dynamic benchmark**  The advantages of large-scale data collection are clear from an evaluation perspective. Including diverse questions in dynamic benchmarks enables more accurate model comparisons. Additionally, when evaluating using datasets from various sources, it becomes possible to assess more generalizable event forecasting abilities. A recent position paper by Paleka et al. (2025a) emphasizes the importance of dynamic benchmarks with sufficient scale and diversity. Furthermore, when dynamic benchmarks contain numerous questions, more questions reach resolution within given timeframes, allowing researchers to obtain model comparison results more rapidly.

If we consider the discussion in Section 4.1.5, evaluation can be conducted even on unresolved questions by utilizing intermediate prediction estimates at intermediate time points (after the question date). This method can increase available data and reduce the time required to achieve statistically significant model comparisons. To effectively utilize these questions, future research should investigate whether intermediate prediction estimates from unresolved questions serve as statistically valid proxies for outcomes and quantify their contribution to evaluation statistical power.

**Evaluation for proprietary models** Dynamic benchmarks play an essential role, particularly in comparing the performance of proprietary LLM agents with built-in search functions or developing forecasting systems based on these models. Deep Research series models with built-in search functions do not allow users to arbitrarily restrict search periods, therefore, for evaluation purposes, these models must make predictions at current time points and await subsequent result verification. Therefore, more rapid and accurate dynamic benchmark evaluation cycles can accelerate model development and improvement processes for such systems.

**Discussion** In summary, advances in large-scale data collection technology enable evaluation methodology improvements beyond simply expanding training data. More comprehensive and rapid model evaluation becomes feasible through data continuously collected from various sources, which will substantially accelerate research and development of proprietary LLM-based event forecasting systems.

# 6 Broader impacts

**Social value of forecasting** Better predictions can lead to better actions. Forecasts have widespread utility: companies evaluate investments by gauging future situations when making investment decisions (Armstrong, 2001); evaluating the impact of diseases like COVID can inform health policy decisions and save lives (Adam, 2020); predicting the impact of policies informs public policy decisions (Hanson, 2013; Rotaru et al., 2022); forecasting can predict public sentiment and opinion trends (Polymarket, 2024); and forecasts on long-term issues like climate change, nuclear security, and AI progress risk are of interest to many (Wang and Karimi, 2024; Kokotajlo et al., 2025). Human forecasting experts and prediction markets aid these pursuits by popularizing prediction and aggregating human intelligence. As society's understanding of superforecasters and prediction markets has increased, the influence that prediction markets have on society has also been growing (Judgment, 2025; Jones, 2024).

**Section overview** By providing predictions quickly and cheaply, enabling scenario simulation and agent integration, and employing training techniques described in Section 4, LLMs could expand the scope of forecasting beyond what human forecasters are currently capable of. Section 6.1 explains the impacts that the predictive abilities of event forecasting AI can directly have on society. Section 6.2 presents ideas and proposals about scenarios where event forecasting capabilities enhance predictive intelligence and simulate the future. Section 6.3 explains key challenges that remain in integrating event forecasting AI into society and proposes technical directions that can overcome these. Section 6.4 explains potential risks associated with event forecasting technology.

## 6.1 Direct social impact of event forecasting AI

**Quantitative expansion of answered event forecasting questions** Event forecasting AI can provide automated answers to various questions not handled by prediction markets, significantly expanding the volume of available predictions to the world (Matthews, 2025). While prediction markets address numerous forecasting questions, a substantially larger set of predictive events remains outside their scope. Furthermore, markets with limited participation may exhibit poor predictive performance (Bosse, 2023c). Also, unlike prediction markets that take time from problem registration to human forecasting when certain events suddenly occur, AI can deliver real-time probability assessments in accessible formats when required.

In addition, AI can address several inherent limitations of prediction markets. For example, for politically and economically important problems where predictions directly influence the real world, prediction markets can be targets of market manipulation. Notably, there have been reports that attempts to manipulate Polymarket probabilities for certain candidate wins in the US presidential election ended in failure (Andrew, 2024). In contrast, AI predictions are relatively free from such direct attacks (however, alternative risks exist, which are described in Section 6.4).

AI can also answer private or personalized queries that are difficult or unsuitable to ask prediction markets or human forecasters. Individuals may be curious about which charitable organization would have the best impact when donating given their personal social good value. They may also be curious about which policies best align with their personal values among those proposed by different political parties. According to our

discussion in Section 3, there is evidence that AI has already surpassed general public prediction performance in several fields. This evidence implies that the current level of AI's prediction already has the potential to be a level that general people can refer to in specific domains.

**Addressing questions without clearly defined resolution conditions**  We further argue that event forecasting AI can help address questions without clearly defined resolution conditions based on various techniques discussed in Section 4. Examples of such questions include the personalized questions mentioned earlier about which charitable organization would be most effective or which policy would be most appropriate. These questions do not have quantitative measures that can be evaluated. However, as mentioned in Section 4.3.3 earlier, since LLMs have the ability to generate other questions related to each question, this capability enables the decomposition of ambiguous questions into well-defined subquestions. For example, the charitable organization question can be reformulated as inquiries about measurable economic and social outcomes resulting from specific donation allocations over defined time periods. As another example, policy evaluation question can be reframed as predictions about economic indicators like GDP that will change after one year or questions predicting survey results after one year. In practice, such conditional predictions align with widespread public interests, and prediction markets operate markets for conditional questions about what changes important policies will bring to observable indicators (Metaculus, 2024). However, if we want to consider every possible combination of conditions and their various implications for conditional predictions, the number of markets that need to be operated becomes prohibitively large, so there are limits to handling many questions scalably in prediction markets. Event forecasting AI can address this limitation.

Meanwhile, we further propose methods for AI to directly answer questions without clearly defined resolution conditions without creating clear questions. Basically, AI can provide values as rough estimates based on its own criteria even for questions without clearly defined resolution conditions, by leveraging predictive capabilities developed for well-defined forecasting questions. Additionally, AI can provide values aligned with learning objectives rather than rough estimates through targeted training. One training method that can be used is using the model's prediction value $q_1$ made at an intermediate time point $t_1$ (after the question date $t_0$ but before resolution date $t_2$), which incorporates additional information unavailable at the original question time, discussed in Section 4.1.5. This enhances the model's capability to answer questions with ambiguous resolution criteria by exploiting information differences between past and future time points. Another training method is applying the method from Section 4.3.3 mentioned earlier. In this method, learning is performed by receiving reward signals from various subquestions with existing resolutions that can be created from questions without clear resolutions.

**Combination of AI and expert predictive intelligence**  Human experts can enhance societal forecasting capabilities by augmenting AI's predictive abilities (Passerini et al., 2025). Human forecasters can simply refer to AI's predicted values for their own predictions, and they can also utilize related search documents and evidence that AI provides together (Halawi et al., 2024). Additionally, experts can provide AI performance by supplying supplementary information beyond the system's access. Alternatively, they can combine AI predictions with human forecasts in ensemble methods for final predictions (Schoenegger et al., 2025). Experts can also selectively utilize AI-predicted values by tailoring them according to specific criteria.

For example, human forecasters can leverage AI for algorithmic trading in markets, thereby AI can contribute to society in this way. AI-assisted trading by human forecasters represents one form of expert-AI collaborative forecasting (Li et al., 2025). Turtel et al. (2025b) demonstrated that they made money in backtests by taking positions when the model's probability estimates exceeded market predictions by a specified threshold. It is important to recognize that since markets are zero-sum games, publicly available superforecaster AI technology will not be able to increase the number of people who gain additional profits. However, by lowering the costs needed for experts to make appropriate predictions, more liquidity is supplied to markets and the qualitative value of predictions that markets provide increases. Companies, groups, or individuals with professional information about finance can leverage AI predictive capabilities to enhance forecasting efficiency in prediction markets and, by extension, traditional financial markets.

## 6.2 Expansion of predictive intelligence

**Conducting future simulation about the world**   We argue that AI has the potential to not only estimate probabilities for individual scenarios but also creatively generate and propose interconnected sequential scenarios, helping individuals and society better understand potential futures (Moreira et al., 2025). Current markets or forecasting experts focus on predicting probabilities for each given question by people. However, LLMs also have strong capabilities for brainstorming events that affect the event and other events that the event affects. Through scenario trees generated by LLMs, users can gain insights about which scenarios are better and what actions achieve better scenarios (Pérez-Ortiz, 2024; Futurist, 2024).

**Integrating predictive intelligence into LLM agents**   We also argue that the development of event forecasting AI can contribute to improving the performance of general LLM agents. When general LLM agents need to solve problems based on uncertain assumptions about society or predictions about technological development, unlike deterministic domains such as mathematics or coding, event forecasting AI capabilities can be seamlessly integrated into the probabilistic components at various stages of the model's reasoning process.

For example, in questions about new technology adoption strategies, rather than providing generic recommendations, LLM agents can propose more practical strategies by quantitatively evaluating market success probabilities of related technologies, potential competitor responses, and regulatory change likelihoods through event forecasting AI.

As another example, an AI scientist (Luo et al., 2025; Gottweis et al., 2025; Yamada et al., 2025; Starace et al., 2025) system could leverage event forecasting capabilities to evaluate the likelihood of experimental success before resource allocation. As discussed in Section 5.3, research paper data can enhance these capabilities. Furthermore, this predictive ability can be enhanced by incorporating data from experimental results, creating a virtuous cycle that accelerates scientific discovery. In the longer term, these predictive abilities could be used to inform the overall direction of research planning.

Several methods enable this integration. One approach involves incorporating event forecasting training within general LLM training procedures. Through multi-task learning that performs event forecasting training and general LLM alignment learning together, predictive reasoning abilities can be naturally integrated into the model's overall reasoning process. This enables more sophisticated judgments even in situations with high uncertainty. Alternatively, a modularized approach enables general LLMs to utilize event forecasting AI using agent2agent (A2A) methods (Google Developers, 2025) or as external world models (Hu and Shu, 2023). In this approach, when LLMs engage in complex reasoning involving uncertain assumptions about the future, they can draw more grounded conclusions by receiving real-time probabilities of related event occurrences or scenario simulations from event forecasting AI.

This integration of predictive intelligence makes AI systems more than conventional information providers, enabling principled probabilistic reasoning about uncertain futures rather than just deterministic logical inference.

## 6.3 Social and technical challenges for social application

For event forecasting AI to actually be used in society, there are additional social and technical barriers that must be overcome beyond prediction performance improvements. The primary challenges include: (1) systematically evaluating prediction reliability in various real situations and (2) effectively conveying this reliability information to users.

**Systematic evaluation of prediction reliability**   From the standpoint of providing model prediction probabilities to actual society, comprehensive understanding of event forecasting performance (e.g., Brier score or ECE) beyond standard benchmark distribution is required. Specifically, systematic research is needed on how performance varies across various dimensions (Paleka et al., 2025a). For example, (1) different domains in real needs, where users' expert knowledge may be superior to AI in some areas while AI excels in others;

(2) temporal distribution shifts, as prediction models trained on past data may not maintain performance over time; and (3) prediction timeframes, distinguishing between short-term and long-term forecasting accuracy.

Additional metrics that interpret AI probability outputs need to be devised. For example, reliability metrics for AI probability estimates can be established (Ulmer et al., 2024). For instance, when both user predictions and AI predictions exist, reliability can be defined as the concept of "how much weight should be given to AI predictions" when users seek to obtain final prediction values. Bayesian perspectives can be applied to assessing AI prediction reliability, enabling users to calibrate their confidence in AI outputs based on historical performance, domain expertise, and uncertainty quantification (Trick et al., 2023). Such methods can provide formal methods for combining human judgment with AI predictions under uncertainty.

**Effective communication of reliability information**   A system interface is needed that enables users to judge whether they can trust the predictions that AI provides (Matthews, 2025; Marusich et al., 2025). Due to the nature of predictions, users cannot readily evaluate AI performance through isolated examination of probability values for individual events. This differs from general LLM agents, which can be validated through knowledge queries with verifiable answers obtained from external sources such as search. Therefore, additional research into methods for allowing users to gauge event forecasting AI's reliability is required.

One method is building a prediction history browsing system. If models continuously make predictions about events across multiple domains in interfaces similar to prediction markets, users can assess the model's reasoning processes and accuracy patterns by analyzing historical predictions within specific domains. When alternative predictions, such as market forecasts, are available for the same events, comparative analysis becomes possible. Unlike prediction markets, which are limited by the need for human participation and economic incentives, AI systems can generate predictions for a much broader range of events at scale, providing users with more comprehensive performance data for evaluation.

Another method is for users to directly compare their own insights with AI's insights by answering various prediction questions themselves. Users can compare their own guesses with various questions that AI has already answered. Alternatively, users can also directly ask AI various questions and compare their responses by matching the probabilities.

Methods to appropriately convey to users whether event forecasting performance is low in specific situations or domains are also required. Models can be made to express low confidence in such answers or decline to predict. Alternatively, models can communicate their confidence levels through well-designed user interface elements that clearly indicate prediction reliability.

## 6.4   Potential Risks of Event Forecasting AI Expansion

Consideration of potential risks of event forecasting AI is essential. As AI systems provide predictive information to users, these predictions influence individual and societal decision-making processes. While current AI predictions have not yet achieved superforecaster-level performance, as AI capabilities approach this threshold, their societal influence will increase proportionally.

**Self-fulfilling predictions and negative consequences of predictions**   Investigation is required into self-fulfilling forecasting that AI creates and its societal consequences (van Amsterdam et al., 2025; Bauer and Gill, 2024). For example, when AI predicts an economic recession and influences investor sentiment, AI has a negative effect on the economy. This is similar to the potential risks that short selling creates in existing traditional financial markets. Therefore, further research is required on self-fulfilling prediction scenarios and their negative impacts.

**Malicious Attacks on Prediction Systems**   Consideration must be given to AI's reliability, bias, and robustness. As discussed earlier, AI is relatively free from intentional trading attempts to change prediction probabilities through markets. However, preparation is needed to address new forms of manipulation attempts targeting AI. For example, adversaries may attempt to manipulate AI predictions by creating or influencing markets with intentional bias, knowing these markets will be incorporated into AI training datasets (Fu et al., 2025). Furthermore, malicious actors may bias AI predictions by strategically publishing misleading content

on the web that AI systems may retrieve during information gathering. AI developers' intentional bias in predictions may also arise, so careful model development, evaluation, and monitoring protocols are essential to detect and prevent intentional manipulation of prediction outcomes.

**Excessive belief in inaccurate predictions**   When models make inaccurate predictions and users place excessive confidence in them, this can cause negative social effects (Klingbeil et al., 2024). In addition, when users experience negative outcomes from overreliance on inaccurate predictions, this erodes public trust in event forecasting AI, potentially hindering future adoption and continued technological development. Users should be enabled to have appropriate levels of belief in models' reliability. This requires research efforts on confidence in forecasting, such as those described in Section 6.3.

**General considerations about AI's impact on society**   Even in situations where AI predictions are accurate to a certain level, how society will accept and utilize AI's predictions is an unknown area. Sociological and technical considerations must be conducted regarding AI having influences on society's overall decisions in general (Kokotajlo et al., 2025).

## 7   Conclusion

We have written a position paper advocating for large-scale training of event forecasting LLMs. We argued that recent technical foundations for event prediction have been established, that additional learning techniques and large-scale dataset construction can enable models to approach superforecaster-level, and that improved event forecasting models will have significant societal impact.

Numerous follow-up studies are needed to advance the event forecasting field. Developing the infrastructure for event forecasting research, including information retrieval systems, needs to be conducted in academia (Section 2.4). The distinctive machine learning characteristics of event forecasting and specialized learning methodologies need to be further explored (Section 4). The identification of diverse event forecasting training datasets and demonstration of multi-source data training approaches will be central to progress in this field (Section 5). Through effective utilization of historical data and scaling law research, we can chart a path toward superforecaster-level performance (Sections 4.2 and 3). Through comprehensive large-scale training, we can validate proposed methods in this field and demonstrate real-world application potential via products and demonstrations (Sections 3 and 6.1). Researchers can participate in existing event forecasting challenges and share their performance results (Williams, 2025b; Karger et al., 2025), or contribute to the advancement of these challenges themselves. Through research that expands predictive capabilities and integrates forecasting into existing AI agent reasoning, we can increase the broader societal impact of prediction-based AI systems (Section 6.2). Research is needed to identify effective applications and address potential concerns regarding event forecasting across domains such as society, finance, healthcare, and policy (Section 6). In summary, the prospects for event forecasting research are promising. We encourage the academic community's engagement and participation in advancing this important field.

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

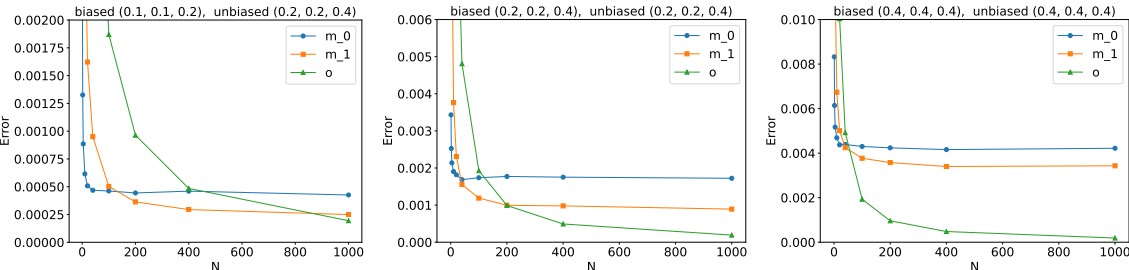

Figure 2: Results of the simulation. The x-axis represents $N$ while the y-axis represents the error between averaged estimates (average of $m_0$, $m_1$, and $m_2$, respectively) and $P_{hidden}$. $N$ represents the simulation trials. The left three numbers in the title represent the biased noise terms for $\alpha$, $\beta$, and $\pi$. The right three numbers represent the unbiased noise terms for $\alpha$, $\beta$, and $\pi$.

## A  Details on hypothetical event Bayesian networks

$m_0$ and $m_1$ are estimates of $P(o = 1|S_0)$ and $P(o = 1|S_1)$, respectively, and are sampled according to six noise term parameters corresponding to the three parameter terms above. Among the six parameters, three represent biased noise for the actual parameters, while the remaining three represent unbiased noise. Biased noise represents the collective and intrinsic error of expert forecasters in optimal situations for parameter probability estimation, while unbiased noise represents sampling error related to which forecasters participated. addNoise refers to noise addition in log probability space.

$$\hat{\alpha} = \text{addNoise}(\alpha, N(\sigma^2_{biased,\alpha}, \sigma^2_{unbiased,\alpha}))$$
$$\hat{\beta} = \text{addNoise}(\beta, N(\sigma^2_{biased,\beta}, \sigma^2_{unbiased,\beta}))$$
$$\hat{\pi} = \text{addNoise}(\pi, N(\sigma^2_{biased,\pi}, \sigma^2_{unbiased,\pi}))$$
$$t \sim \pi$$
$$m_0 \sim (1 - \hat{\pi}) \cdot \hat{\alpha} + \hat{\pi} \cdot \hat{\beta}$$
$$m_1 \sim (1 - t) \cdot \hat{\alpha} + t \cdot \hat{\beta}$$
$$o \sim (1 - t) \cdot \alpha + t \cdot \beta$$

Thus, in this modeling, $m_0$ and $m_1$ can be considered as values obtained through optimal inference taking into account noise with respect to $\alpha$, $\beta$, $\pi$.

Experimental results are represented in Figure 2. We present graphs showing average errors according to $N$ for three different noise patterns. Specifically, errors are calculated by Brier score, and $\frac{1}{N} \sum_{n=1}^{N} m_{0,n}$, $\frac{1}{N} \sum_{n=1}^{N} m_{1,n}$, and $\frac{1}{N} \sum_{n=1}^{N} o_n$ are used to estimate $P_{hidden}$, respectively.

Figure 2 demonstrates a bias-variance tradeoff between different estimators of the hidden probability $P_{hidden}$. $m_0$ (market predictions at question time) has the lowest variance but highest bias, $m_1$ (intermediate market predictions) falls in the middle, and $o$ (final outcomes) has the highest variance but lowest bias. For small amounts of data (low $N$), market predictions ($m_0$) provide more reliable estimates due to their lower variance. As the amount of data increases, it becomes advantageous to use data closer to the actual outcome ($o$). As bias in market predictions increases, the threshold where training directly on outcomes becomes optimal decreases.

Notably, $m_1$ can be a superior estimate of $P_{hidden}$ under certain conditions, particularly when there is significant uncertainty in the transition from $S_0$ to $S_1$ (high estimation error for $\pi$, as shown in the first panel) and when we have a moderate number of observations $N$. The parameter values were chosen to illustrate different noise scenarios: varying levels of bias vs. unbiased noise, and different uncertainty levels for the intermediate state transition ($\pi$).

