# OpenReview forum: "Advancing Event Forecasting through Massive Training of Large Language Models: Challenges, Solutions, and Broader Impacts"
_TMLR — Rejected by TMLR_

### Review · Reviewer_yZgT · 2025-10-23

**Summary Of Contributions:**

The paper argues that it is the time to massively train superforecaster‑level event‑forecasting LLMs and presents a clear roadmap: methodologically, it formalizes three core challenges (noisiness and sparsity, knowledge cut‑off, simple rewards) and proposes a hypothetical event Bayesian network to unify labels and dependencies, leveraging poorly recalled and counterfactual events to expand signals, and auxiliary rewards to enforce process quality and calibration; on the evaluation side, it advocates a tri‑source strategy (markets, public structured indicators, crawled news) and discusses applications and governance.

Key strengths include that:
- Truly trustworthy and accurate forecasts would be impactful.
- The paper lays out clear and practical ways to fix the main problems it identifies, resulting in an actionable roadmap.

However, several major weaknesses remain:
- Hallucination is a recognized concern that is not discussed in a systematic way. However, hallucination is not fully discussed in this paper.
- In domains demanding high trust like finance and healthcare, forecasts by LLMs may still be hard to accept and adopt, even with scores and intervals, which limits real‑world use if not addressed.
- The core proposals remain untested in task settings. Section 4 outlines the event BN, counterfactual and low‑recall augmentation, and auxiliary rewards, but there are no end‑to‑end experiments or ablations that could support these proposals.
- On the evaluation side, the main new element is a generic LLM‑as‑a‑judge for process assessment. Beyond that, existing metrics are largely reused, and there is no forecasting‑specific or unified composite score, which limits the novelty and structure of the evaluation.

**Additional Comments:**

NA

**Audience:**

Yes

**Audience Explanation:**

Researchers interested in real-world forecasting in operations, public policy, and finance will find the paper's insights interesting. Its roadmap provides useful advice for developing, evaluating, and implementing forecasting systems, with practical implications for audiences in academia and business.

**Broader Impact Concerns:**

Forecasting systems can feed back into the world, creating self‑fulfilling or self‑defeating effects that alter the very events being predicted. They are vulnerable to misuse and manipulation, including data poisoning and benchmark gaming, which can mislead decision makers in settings with high trust requirements. Unequal access to models and data may also increase inequity and distort markets.

**Claims And Evidence:**

No

**Claims Explanation:**

This is a position paper, so several central claims are framed as directions rather than demonstrated results. While some formal scaffolding is outlined—for example, a Bayesian framing for hypothetical event dependencies—the key proposals like leveraging poorly recalled and counterfactual events, auxiliary rewards, and a tri‑source, contamination‑aware evaluation pipeline are not supported by task‑specific, reproducible experiments. As written, the evidence is insufficient to support some key claims.

**Requested Changes:**

Strengthen empirical grounding and reproducibility, like adding minimal in-domain cases with clean evaluation that directly demonstrate gains and robustness for the three proposals, with clear baselines and simple reproduction, and then the path from idea to usable method is evident.

Explain hallucination and its impact in this forecasting setting and its common triggers, show a few representative failures, and explain how they distort evidence and confidence and ultimately shift predictions and downstream decisions.

Provide a path about how to use LLMs to forecast in domains demanding high trust like finance and healthcare by outlining a simple path that makes predictions credible and including one brief end‑to‑end example.

Beyond LLM‑as‑a‑judge, introduce genuinely new metrics or a single composite score that unifies accuracy, calibration, and horizon resolution to improve novelty and bring structure to the evaluation. This replaces a list of scores with a coherent, comparable benchmark across papers and datasets.

---

### Review · Reviewer_kSps · 2025-10-25

**Summary Of Contributions:**

This position paper argues that recent progress in large language models (LLMs), reasoning models, and reinforcement learning makes large-scale training for superforecaster-level event forecasting feasible. It identifies key training challenges unique to this domain 爱你的proposes methodological solutions to strengthen reasoning and generalization. The paper also outlines strategies for large-scale data acquisition from market, public, and web-crawled sources to support training and dynamic benchmarking. This paper also provide discussion on the broader societal implications of LLM-based forecasting, highlighting opportunities for predictive intelligence and challenges such as evaluation integrity and misuse risks. As a position paper, it provides a valuable summary of current methods and proposes promising future directions for event forecasting with LLMs. However, due to its limited discussion on the empirical performance of existing methods, the limitations and practical validity of the proposed future directions remain uncertain.

**Audience:**

Yes

**Audience Explanation:**

The paper addresses the application of LLM in advancing event forecasting, which is a very interesting topic. Its comprehensive summary of current challenges, proposed methodological innovations, and exploration of societal implications provide valuable insights for researchers in machine learning, AI alignment, and applied forecasting. Given the growing academic and practical interest in superforecaster-level AI systems, this paper would attract attention from both theoretical and applied audiences.

**Broader Impact Concerns:**

The use public, market, and web-crawled datasets introduces risks related to consent, privacy, and intellectual property, especially when using data scraped from news, social media, or web sources without explicit permission.

**Claims And Evidence:**

No

**Claims Explanation:**

The paper makes several strong and ambitious claims regarding the feasibility and potential of large-scale training for superforecaster-level event forecasting. However, these claims are not fully supported by clear or convincing empirical evidence. As a position paper, it mainly summarizes prior work and outlines conceptual directions rather than presenting systematic experiments or quantitative validation. The discussion of existing model performance and concrete comparative results is limited, making it difficult to assess how well the proposed methods address current limitations in practice.

**Requested Changes:**

To strengthen the contribution and demonstrate practical feasibility, empirical validation is necessary.Specifically, the authors should summarize and compare empirical results from existing works to support the proposed training ideas

---

### Review · Reviewer_aARt · 2025-10-30

**Summary Of Contributions:**

The paper presents a timely and structured position on advancing event forecasting using large-scale training of LLMs, arguing that recent progress in reasoning models and reinforcement learning makes superforecaster-level AI feasible. It identifies key challenges: noisiness and sparsity, knowledge cut-off, and simple reward structures, and proposes coherent mitigation strategies such as hypothetical Bayesian modeling, counterfactual training, and diverse data sourcing.

Strengths:

- Comprehensive and well-organized discussion of current challenges and research directions.
- Clearly articulated motivation linking LLM evolution to forecasting performance.
- Innovative proposals (e.g., hypothetical event Bayesian networks, auxiliary rewards).
- Strong grounding in recent literature and benchmarks, showing deep field awareness.

Weaknesses:

- Primarily a conceptual or position paper, lacks robust empirical validation.
- Some proposals (e.g., large-scale crawling, counterfactual training) are ambitious without detailing feasibility or ethical implications.
- Overreliance on existing LLM advancements without demonstrating concrete experimental pathways.
- Repetitive framing across sections could be condensed for clarity

**Additional Comments:**

I simple don't see this paper to be sufficient for publication at this stage.

**Audience:**

No

**Audience Explanation:**

it simply lacks sufficient empirical evidence.

**Broader Impact Concerns:**

-

**Claims And Evidence:**

No

**Claims Explanation:**

While the paper provides a thorough conceptual argument and cites relevant recent studies, it lacks robust direct empirical evidence or experiments to substantiate its claims. Most assertions are speculative or based on secondary references rather than demonstrated results, making the evidence suggestive but not convincingly conclusive.

**Requested Changes:**

I would suggest to carry out the diverse methods they propose, which are extensive.

---

### Author Response · Authors · 2025-11-13
**Author Response: Position-Paper Intent and Next Steps**

We sincerely thank the reviewers for their thoughtful and detailed feedback.


Our intention in submitting this work was to present it as a position paper: a conceptual and discussion-oriented piece rather than an empirical study. Several papers published in TMLR have taken a similar format without experimental components (e.g., Open Problems in Technical AI Governance, TMLR, 2024), and we understood that TMLR welcomes rigorous reviews of such position papers.


If experimental validation of the proposed ideas is considered essential for journal publication, we believe that such empirical work would be better developed as a separate follow-up paper focusing on implementation and verification. Conversely, if emphasizing the review-oriented and conceptual aspects improves the paper’s quality and suitability for TMLR, we would be glad to revise in that direction. In this work, the experimental suggestions primarily serve as devices to illustrate current challenges in the field rather than as concrete empirical claims.


Interpreting the reviewers’ feedback from the perspective of a position paper, we understand that the current version may not yet fully meet TMLR’s standards for publication in this category. Nonetheless, if the reviewers believe the paper could gain value by toning down or removing experimental elements and strengthening its analytical or survey components, we are open to that revision path.


Several of the questions raised in the paper have seen notable developments in the last three months.
New major event-forecasting benchmarks have been introduced ([link](https://arxiv.org/abs/2508.11987), [link](https://arxiv.org/abs/2510.17638), [link](https://forecastingresearch.substack.com/p/put-your-ai-forecaster-to-the-test)).
Some recent reports indicate that, from a calibration perspective, model forecasts can now outperform market baselines.
The size and complexity of datasets in this field have been increasing.
A report showed that large-scale training enables models to generate measurable market gains ([link](https://blog.lightningrod.ai/p/foresight-32b-beats-frontier-llms-on-live-polymarket-predictions)).
We plan to incorporate these updates into the next version of the manuscript, as they may directly address parts of the reviewers’ questions.

---

### Decision · Action_Editor_7dVM · 2025-12-11

**Recommendation:** Reject

**Audience:**

No

**Audience Explanation:**

The research community for forecasting, especially event forecasting, will be interested in the topics of this work. However, the work needs further development, as suggested by the reviewers, to address the many weaknesses of LLM-based solutions and to highlight the feasible pathways forward.

**Claims And Evidence:**

No

**Claims Explanation:**

All reviewers agree that the position paper does not fulfil the goals it targets, and recommend to reject unanimously. In the original reviews, the strengths are summarized while the weaknesses are highlighted. In the response letter, authors have acknowledged that several concerns are valid but will address them in a future version. That said, authors did not provided a revised version.